# Solar energy and regional coordination as a feasible alternative to large hydropower in Southeast Asia

Kais Siala[1], Afm Kamal Chowdhury [2,3], Thanh Duc Dang [3] & Stefano Galelli [3✉]

Strategic dam planning and the deployment of decentralized renewable technologies are two elements of the same problem, yet normally addressed in isolation. Here, we show that an integrated view of the power system capacity expansion problem could have transformative effects for Southeast Asia's hydropower plans. We demonstrate that Thailand, Laos, and Cambodia have tangible opportunities for meeting projected electricity demand and $CO_2$ emission targets with less hydropower than currently planned—options range from halting the construction of all dams in the Lower Mekong to building 82% of the planned ones. The key enabling strategies for these options to succeed are solar PV and regional coordination, expressed in the form of centralized planning and cross-border power trading. The alternative expansion plans would slightly increase the cumulative costs (up to 2.4%), but substantially limit the fragmentation of additional river reaches, thereby offering more sustainable pathways for the Mekong's ecosystems and riparian people.

[1] TUMCREATE Ltd., Singapore, Singapore. [2] Environmental Studies Department, University of California Santa Barbara, Santa Barbara, CA, USA. [3] Pillar of Engineering Systems and Design, Singapore University of Technology and Design, Singapore, Singapore. ✉email: stefano_galelli@sutd.edu.sg

I n many developing regions, economic growth is supported by power systems that rely on cheap and locally available energy sources. Southeast Asia is no exception: the region is on the way to achieve universal access to electricity by largely banking on hydroelectricity and fossil fuels[1]. Aside from $CO_2$ emissions, a major concern for this energy policy is the socio-environmental externalities of hydropower development. The main center of activity has been the Mekong River, a global hotspot of biodiversity and home to the world's largest freshwater fishery[2,3]. The Mekong and its tributaries have abundant hydropower potential, part of which has so far been developed by China, Laos, Thailand, and Vietnam—with Cambodia and Myanmar playing a marginal role. Up to 2020, the combined installed capacity of all commissioned dams (>1 MW) is about 41.6 GW[4-6], including some particularly controversial dams recently built on the main stem of the Lower Mekong (e.g., Nuozhadu: 5850 MW, Xayaburi: 1285 MW, Don Sahong: 260 MW). Their impact is profound: by creating artificial storages and fragmenting the river network, they not only alter the hydrological regimes[7,8] but also block fish passage and reduce the transport of sediment and nutrients[9], ultimately affecting the riverine ecosystems, its fisheries, and the

riparian communities[10-12]. If all proceed as planned, another 22 GW will be deployed in the next decades[4-6] (Fig. 1a). And yet, there might be multiple alternatives to this plan: the availability of regional grid interconnections[13,14] and renewable energy sources, particularly solar photovoltaic (PV)[15], suggests that dam development may be partially offset by deploying renewable technologies in low-cost and well-accessible areas[16]. A complementary strategy is the design of more sustainable dam portfolios[17]. Whether any of these alternatives is technically feasible and economically reasonable for it to succeed remains an open question.

Designing dam portfolios and deploying decentralized renewable technologies are two elements of the same problem—i.e., planning the expansion and operations of sustainable power systems—but they are normally addressed with different tools. Strategic dam planning typically relies on multi-objective optimization frameworks that balance hydropower capacity with one, or multiple, environmental objectives, such as fish biomass and biodiversity losses[18], sediment supply[19,20], or greenhouse gas emissions from reservoirs[21]. These studies provide fundamental guidance for future hydropower projects, as they identify

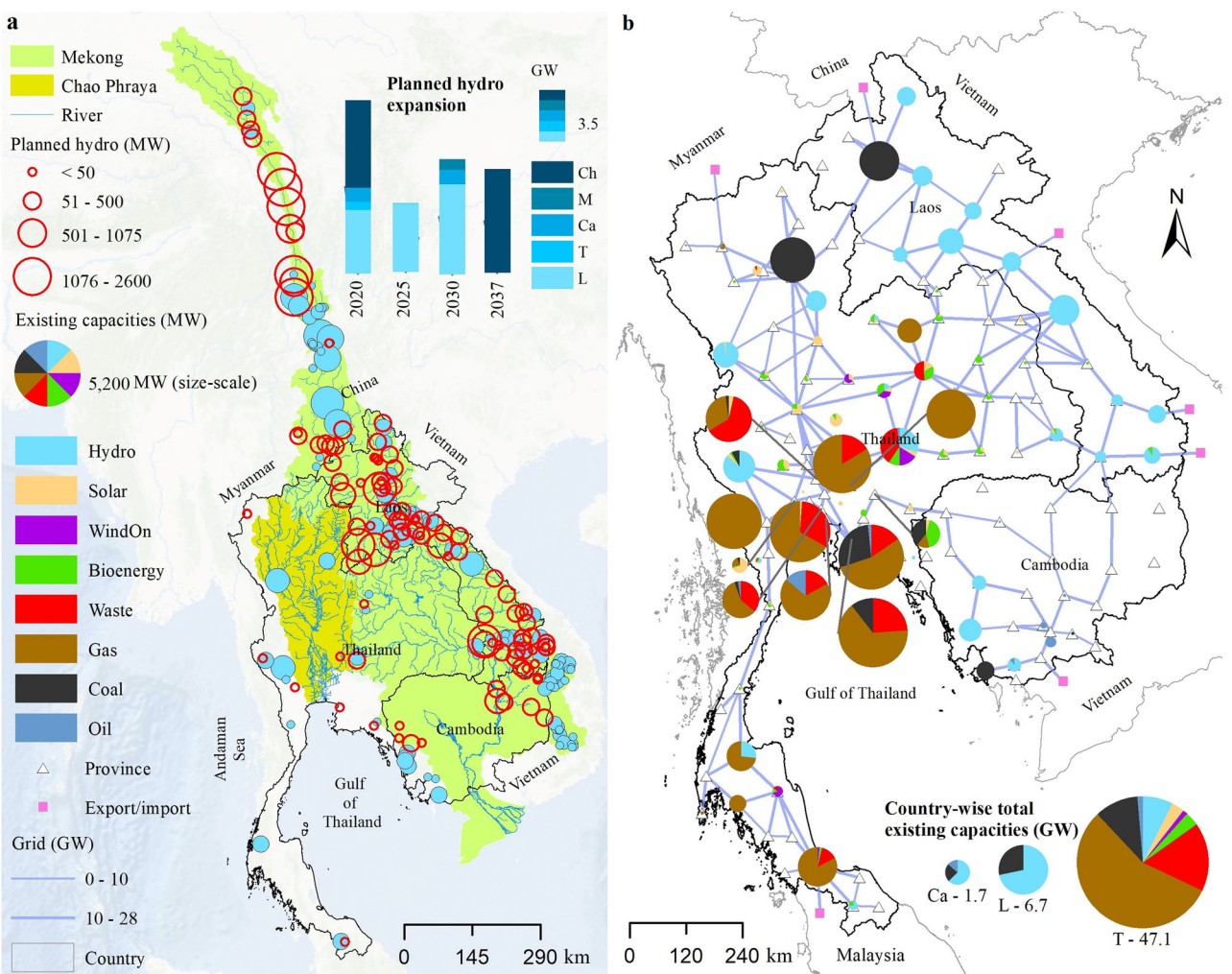

**Fig. 1 Study site. a** Full spatial extent of the Chao Phraya and Mekong basins, together with the dams operated in 2016 (blue-filled dots) and planned (red circles) by all riparian countries. These dams are modeled by the hydrologic–hydraulic model VIC-Res. The bar chart in top right corner shows planned hydropower capacities over the period 2020–2037, with color scale indicating riparian countries (L Laos, T Thailand, Ca Cambodia, M Myanmar, and Ch China). **b** Spatial representation of the power system infrastructure for each province of Thailand, Laos, and Cambodia. Segments, triangles, and squares represent high-voltage transmission lines, provinces, and import/export nodes. The pie charts overlapping the provinces indicate their existing generation capacities, proportionate to the size scale shown in the left panel. The pie charts at bottom right corner show the country-wise total existing capacities. All components of the power grid were operational in 2016.

opportunities for better trade-offs between power supply and ecosystem services. Most importantly, they highlight the necessity of regional coordination in dam planning[20], as opposed to the piecemeal approach adopted in many basins. However, some dam portfolios may be technically or economically unfeasible, because the frameworks with which they are designed do not represent the role played by dams within power systems. For example, concentrating dams within a few sub-basins may provide an opportunity to balance installed capacity with ecosystem services, but such a plan may be impaired by the cost of developing an adequate transmission infrastructure[22] or if the intermittent production of hydropower dams cannot be absorbed by the existing thermoelectric facilities[23]. These are the fundamental mechanisms captured by the tools used to study the integration of renewable technologies within existing grids, which combine long-term capacity expansion and detailed power system operations[24–26]. The flipside here is that power system planning models typically forgo the information on socio-environmental externalities available from dam planning studies[27] and use simplistic representations of hydropower storage dynamics[28], thereby neglecting hydropower response to climate variability as well as the cascading effect of hydropower operations in large reservoir networks, such as the one being developed in the Mekong.

Here, we introduce a modeling framework for dam and power system planning in the Lower Mekong River Basin that brings the aforementioned elements under the same umbrella. Our framework consists of two components, urbs[29] and the Variable Infiltration Capacity (VIC)-Res model[30,31]. urbs co-optimizes capacity expansion for generation, transmission, and storage as well as hourly power system operations—thus accounting for the balancing of supply and demand, transmission constraints, ramping limits, electricity reserve, and the time needed to start-up and shut down the thermoelectric units. A fundamental feature of urbs is its spatially distributed nature: the model explicitly accounts for the power system infrastructure of 120 provinces in Laos, Thailand, and Cambodia, where a cross-border, power-trade infrastructure is already in place (Fig. 1b). Thanks to this set-up, urbs integrates complex weather data and characterizes the spatial variability of renewable energy sources and hydropower, a fundamental requirement for large-scale studies (cf. ref. [24]). The hydropower availability of each existing and planned dam in the Mekong is calculated using VIC-Res, a spatially distributed hydrologic–hydraulic model simulating not only the relationship between hydro-meteorological forcings and water availability through the basin but also the storage dynamics and turbine release of each reservoir. VIC-Res is also implemented for the Chao Phraya, the second main basin of our study site and home to a few large dams feeding the Lower Mekong power grid.

By running our framework over the period 2016–2037, we show that the regional electricity demand and $CO_2$ emission targets can be met by constructing only 82% of the planned dams in Thailand, Laos, and Cambodia. The key enabling technologies for this alternative to succeed are solar PV and high-voltage transmission lines, which redistribute cheap electricity across distant load centers. Our analysis of alternative dam portfolios proposes other, more sustainable, options: a careful expansion of the power system could even absorb the halting of the construction of all dams in the Lower Mekong, at a cost of about 10 billion US$ over the period 2016–2037. Finally, we show that the alternative dam portfolios could substantially limit the fragmentation of additional river reaches. However, further alterations of the natural flow regime will depend on decisions made in both Upper and Lower Mekong, thus highlighting the need for multi-sector cooperation efforts between all riparian countries.

## Results

**Capacity expansion plans.** We perform a power system optimization of the Lower Mekong region that takes into account the existing power infrastructure, the projected costs of technologies, as well as future electricity demand and emissions reduction targets. The power systems of the Lower Mekong River Basin face two challenges: meeting the growing electricity demand (projected yearly growth rates are 4.3% for Thailand, 8.8% in Cambodia, and 9.5% in Laos) and decreasing the carbon emissions intensity from an estimated 0.536 t $CO_2$/MWh to a target of 0.308 t $CO_2$/MWh. There are many decarbonization pathways to reach these targets, but they roughly fit into two categories. The first one focuses on shifting from coal to gas (which has a lower carbon intensity), with a moderate expansion of renewable energy technologies. The second relies on a large expansion of renewable energy and a moderate expansion of gas power plants, so that the system can accommodate a continuous usage of coal. The results of the optimization, with regard to the energy mixes of the three countries, reflect a combination of these pathways (Fig. 2). In the short term, due to the stringent assumption on the overall emission intensity, we observe that gas replaces part of the coal generation in Thailand and reduces its dependence on imports from Laos. Gas is the cost-efficient solution because the hydropower dams going into operation in 2020 are not sufficient to reduce the carbon emissions in accordance with the stringent targets for that year, and because the installation cost of solar PV is still relatively high (see Table S1 for an overview of the technology cost assumptions).

The decarbonization strategy shifts drastically from 2025 onwards. The usage of coal is on par with 2016 levels, the relative share of gas decreases, while a huge expansion of renewable energy technologies takes place in the Lower Mekong countries. Since the wind potential is rather limited in the region, the three countries increase the capacities of solar PV (particularly in Thailand) and hydropower (mostly in Laos and Cambodia). Solar PV capacity expansion amounts to 52 GW in 2025 and continues to grow steadily in the following years to reach 68.2 GW by 2037. Thailand alone witnesses an addition of 49.8 GW of solar capacity, which is equivalent to about 42% of its total capacity in 2037. Meanwhile, the hydropower capacity in the three countries increases from 9.3 GW in 2016 to 22.8 GW in 2037. Most of the new capacities are added in Laos (+12 GW),

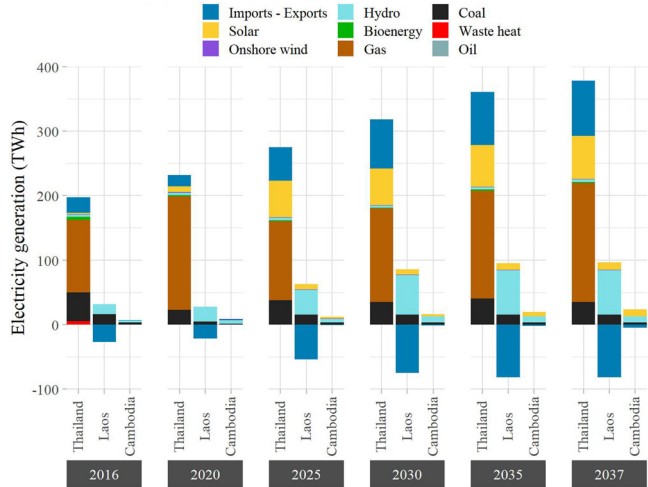

**Fig. 2 Capacity expansion plans for the period 2016–2037.** Evolution of the power mix in Thailand, Laos, and Cambodia designed by urbs. Negative values indicate that electricity exports exceed imports. Note that most of the electricity exported from Laos goes to Thailand.

followed by Cambodia (+1.8 GW), with an additional 0.7 GW in Myanmar dedicated to the Thai power market. This corresponds to an execution rate of 82%, since the total capacity of all planned dams in the region amounts to 17.6 GW. In order to connect the hydropower dams with the demand centers, which are mainly located in Thailand, the power grid is upgraded with the addition of 25 GW bidirectional transmission lines. Consequently, the share of carbon-free generation increases from 16.7% in 2016 to 42.9% in 2037. Whereas hydropower makes up the lion's share in 2016, it only accounts for less than half of the carbon-free generation in 2037. The rest is provided by solar PV (89.2 TWh, or 49.8%), with bioenergy and onshore wind playing minor roles.

**Regional balancing of supply and demand**. The new dams are mainly located in Southern Laos and Northeastern Cambodia, Northern Laos, and Eastern Myanmar. Among the dams that are not selected for the capacity expansion, one is located in Western Cambodia (100 MW) and another one in Southern Laos (70 MW), but the majority (21 dams, about 2.1 GW) are in

Northern Laos. Of the new solar PV capacities, 25% are concentrated in the north west of Thailand, whereas the rest is distributed all over the region. However, most of the power demand occurs around Bangkok. Hence, we observe that different provinces within the three countries play different roles—notably as hydropower generation hubs, solar PV generation hubs, or power demand hubs (the regional distributions of hydro capacities, solar capacities, demand, and transmission lines are shown in Fig. 3). In order to alleviate the regional discrepancies between supply and demand, the model expands the transmission grid in the east–south direction (from Laos to Thailand through Cambodia) and west–south direction (within Thailand), so that most new lines converge toward Bangkok and its surroundings. This cost-optimal power system design implies a high level of regional coordination between the grid operators of the three countries.

**Impact of alternative dam portfolios on power system expansion**. Our results indicate that not all planned hydropower dams must necessarily be built. Moreover, the availability of a vast solar

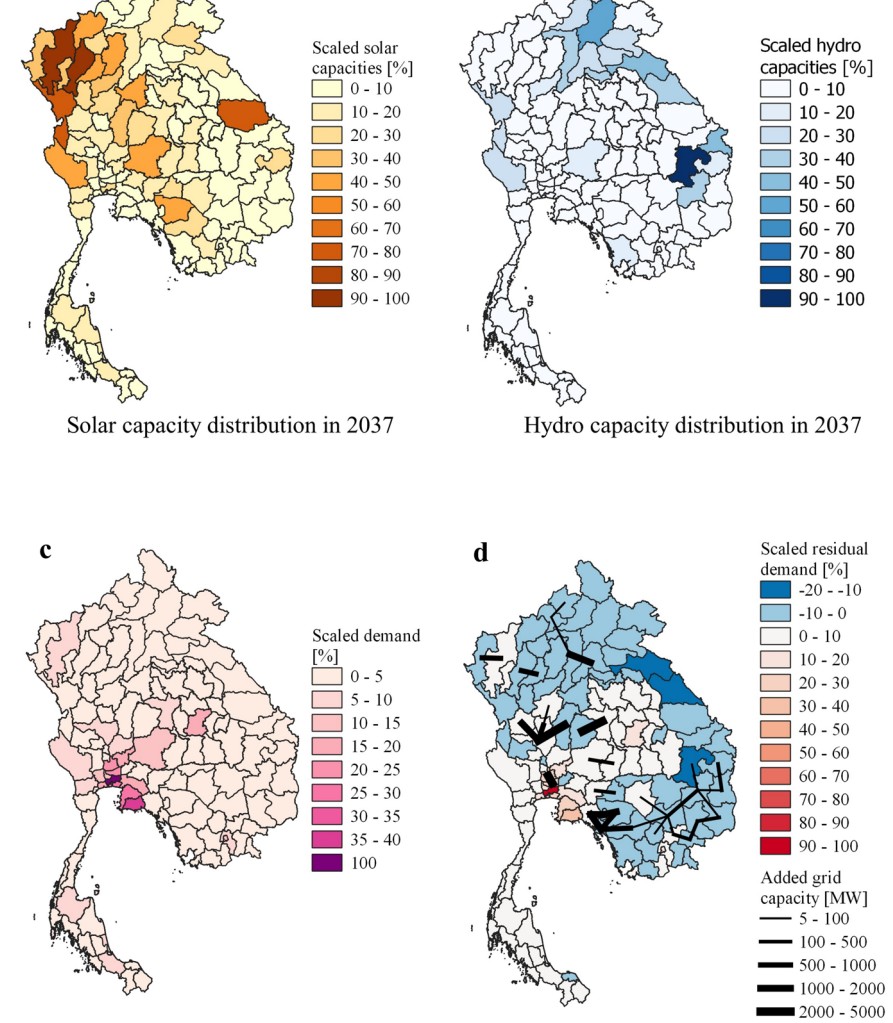

Solar capacity distribution in 2037

Hydro capacity distribution in 2037

Power demand distribution in 2037

Residual demand distribution in 2037 and added grid capacities

**Fig. 3 Spatial analysis.** Regional distribution of the solar capacities (**a**), hydro capacities (**b**), power demand (**c**), and residual energy demand (**d**) in 2037. The maps are scaled to the respective maximum regional quantity. The residual energy demand is calculated by subtracting the annual energy production of solar PV, onshore wind, and hydro from the annual power demand and is indicative of the regional discrepancies between supply and demand. The transmission capacities that are added between 2020 and 2037 are also plotted on subfigure in (**d**).

**Table 1 Dam development portfolios over the planning horizon 2016–2037.**

| Name | Description | No. of dams | | Total capacity (GW) | |
|---|---|---|---|---|---|
| | | L-T-C | Mekong | L-T-C | Mekong |
| Reference | Build all dams (business as usual) | 146 | 236 | 28.46 | 63.63 |
| Stop-Main | Build dams on tributaries only | 139 | 229 | 20.61 | 55.79 |
| Stop-Planned | Stop building all planned dams (from 2020 onwards) | 119 | 203 | 19.24 | 53.70 |
| Stop-All | Stop building all planned and under construction dams (from 2020 onwards) | 111 | 195 | 18.21 | 52.67 |
| 2016 | Dams commissioned by the end of 2016 | 53 | 108 | 9.29 | 29.58 |

Overview of the total number of dams, and corresponding installed capacity, for the business-as-usual plan (Reference) plus three additional dam development portfolios. The last row reports information on dams operational in 2016. Both number of dams and total capacity are reported for all countries falling within the Mekong basin and for Thailand, Laos, and Cambodia only (L-T-C). Data were retrieved from refs. [4–6,35] and processed to filter dams that have either storage capacity >0.1 Mm$^3$ or installed capacity >1 MW.

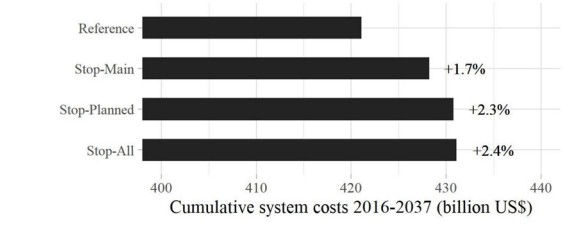

**Fig. 4 Future generation mixes and costs. a** Evolution of the power mix (aggregated across Thailand, Laos, and Cambodia) for the dam development portfolios outlined in Table 1. In **b**, we report the corresponding total system costs.

PV potential[15] suggests that there might be opportunities for further reducing the number of dams built in the near future. We therefore consider three alternative dam portfolios (Table 1) and use urbs–VIC-Res to identify possible substitutes in the power system and quantify the implications in terms of system costs. Two portfolios represent scenarios in which we stop the construction of all dams (Stop-All) or only the planned ones (Stop-Planned)—for which construction works have not started yet. The third portfolio blocks the construction of dams in the main stem of the Mekong (Stop-Main), which have a larger impact on migratory fish populations and sediment supply[20,32].

As illustrated in Fig. 4, the alternative dam portfolios are technically feasible, meaning that a decrease in hydropower production can be offset by other sources, mainly solar PV and gas. Interestingly, there is also a positive correlation between hydropower and coal generation. In fact, if the hydropower share

is high, then the overall carbon-neutral generation is also high. This leaves some freedom to use coal, which is cheaper than gas but has a higher carbon intensity per unit of energy. On the other hand, in the scenarios with less hydropower, the power system has to generate more energy from carbon-emitting technologies without violating the total $CO_2$ constraint, so it resorts to using more gas-fired power plants. Importantly, the alternative portfolios may also be economically feasible: taking into account the investment costs, fuel costs, and fix and variable operation and maintenance costs (up until 2037), our results show that the scenarios with alternative dam portfolios are marginally more expensive than the Reference one. For example, the most restrictive portfolio (Stop-All) leads to cumulative costs that are 2.4% higher than the business-as-usual strategy. This corresponds to about 10 billion US$ over the period 2016–2037.

**Future pathways of river fragmentation and flow regulation.** To estimate and synthesize the combined effects of the alternative dam portfolios on the Mekong's ecosystems, we use the River Fragmentation Index (RFI) and River Regulation Index (RRI)[32,33]. The former captures the effect of dams on the natural connectivity of riverine systems, focusing in particular on long-itudinal connectivity, important for its relation to species migration[18]. The latter quantifies the impact of dams on timing and magnitude of flows: alterations of the natural flow regime that can disrupt the life cycle of freshwater species[34]. When calculating both indices (see "Methods"), we account for the dams selected by urbs for each portfolio and time slice but assume that all dams in China will be constructed as planned—thereby reflecting the lack of coordination between Lower and Upper Mekong countries on infrastructure development.

The RFI and RRI values over the period 2016–2037 give us a glimpse of past, present, and future pathways of river fragmentation and flow regulation (Fig. 5). Dams operational in 2016 appear to affect more the total network regulation rather than the river connectivity, a result explained by two facts. First, most of these dams are located in headwater streams. Second, some of these dams, particularly those in the Upper Mekong, have massive storage capacity (e.g., Xiaowan Dam: 15,043 Mm$^3$, Nuozhadu Dam: 21,749 Mm$^3$), so their effect on the flow regime is perceived across the entire basin[8]. The sudden change in the RFI experienced between 2016 and 2020 (from 17.4 to 66.6%) is largely attributable to dams built in the Lower Mekong Basin, either on the main stem (e.g., Xayaburi Dam, in Laos) or on major tributaries (e.g., Lower Sesan 2 Dam, in Cambodia), which disconnect large fractions of the river network (see Fig. 6). Although the current situation is clearly critical, our results indicate that the alternative dam portfolios could substantially limit the fragmentation of additional river reaches. More precisely, we estimate that future values of the RFI could vary between 66.6

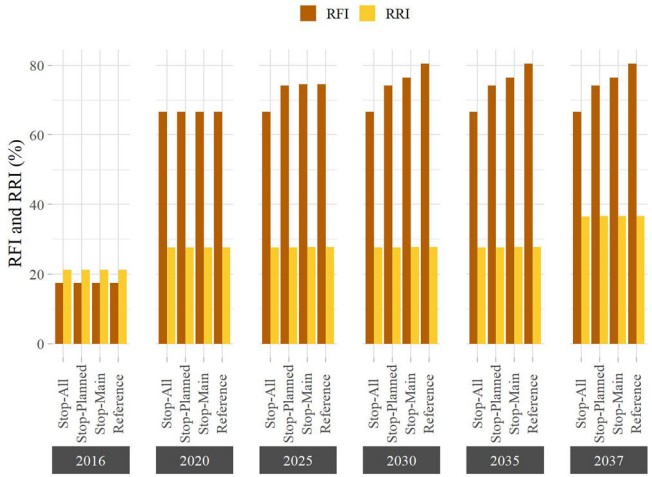

**Fig. 5 Future pathways for the Mekong River Basin.** The figure illustrates the evolution of the River Fragmentation Index (RFI) and River Regulation Index (RRI) between 2016 and 2037 for four different dam portfolios. When calculating both indices, we included for all scenarios the dams planned in the entire basin.

and 80.5%, under the Stop-All and Reference scenario, respectively. Results also suggest that the portfolios have little influence on future alterations of the flow regime, because the dams planned for the Lower Mekong have in general limited capacity to control flows—that is, the ratio between storage and average inflow is small. Instead, the projected RRI increase is mainly attributable to the construction of large-storage dams in the Upper Mekong (see the RRI increase for the year 2037 illustrated in Fig. 5).

A more nuanced understanding of the effect of existing and planned dams on flow regulation is offered by Fig. 6, where we illustrate the Degree of Regulation (DOR), a spatially disaggregated version of the RRI calculated for each river reach (see "Methods"). By contrasting the DOR calculated for the situation in 2020 and the Stop-All and Reference portfolios for 2037, Fig. 6 reveals the differential impact of Lower and Upper Mekong dams on river flows (see Fig. S2 for the DOR values of the other two portfolios). First, the construction of just a few, large-storage dams in China would further alter the flow regime far downstream along the river network (cf. the current situation, 2020, against the Stop-All portfolio). In particular, the DOR would be >25% in almost the entire main stem. To put this number into perspective, consider that Lehner et al.[35] marked the possibility of substantial changes in the natural flow regime for DOR values >10%. Second, the construction of more dams in the Lower Mekong countries would not dramatically change the DOR values in the river network (cf. Stop-All and Reference portfolios), since most dams—even those planned for the main stem—have limited storage capacity in relation to the river flow. In addition, the flow regulation effect is diluted by the presence of a few weakly regulated tributaries, such as those in the southwest part of the basin, primarily controlled for irrigation.

In sum, it appears that the fate of the Mekong's ecosystems is caught between the dam development plans for the upper and lower portions of the basin. Our analysis shows that a careful expansion of the power system in Thailand, Laos, and Cambodia could prevent additional damages on the river's natural connectivity, but future alterations of the natural flow regime are more directly related to the construction of large dams in the Upper Mekong.

## Discussion

Our study demonstrates that Thailand, Laos, and Cambodia could meet their future electricity demand and $CO_2$ emission

targets with substantially less hydropower than what is currently planned—options range from halting the construction of all dams in the Lower Mekong to building 82% of the planned ones. Importantly, the options we explored are both economically and technically feasible. Beginning with the economic aspects, note that even the most restrictive dam portfolio we considered (i.e., halting of the construction of all dams in the Lower Mekong) would increase the cumulative costs over the period 2016–2037 by only 2.4% (~10 billion US$) with respect to the business-as-usual strategy. And while these figures may change in the future in response to cost overruns in large hydropower dams[36] or fluctuations in the cost of technology and commodities, we note that they are comparable to the estimated damages of dam developments on the inland fishing industry alone (i.e., 2–13 billion US$[37,38]). But hydropower dams have many other negative impacts, such as greenhouse gas emissions[39], thermal pollution[40], or the displacement of indigenous communities[41]. In this regard, it is important to consider that several socio-environmental externalities are related to the natural connectivity of the river network[32], meaning that the Lower Mekong countries still have a chance to curb an already critical situation. The flipside of our results is that alterations of the natural flow regime—a potent driver of biodiversity[42]—are also determined by dam planning decisions in the Upper Mekong. China's recent decision to share year-round water data with the downstream countries is a first important step[43], which should ideally be followed by mechanisms for jointly planning infrastructure investments[20].

The reason behind the technical feasibility of these plans lies in the flexibility of the other technologies. Solar PV modules, while subject to a diurnal cycle and to weather conditions, have the advantage of being scalable and deployable in any province of the Mekong countries. In particular, they can be built in every province, spare the costs of long transmission lines, and ensure a higher level of energy autarky. The seasonal fluctuations are low and complement very well the existing hydropower production, provided that there is a strong coordination between the national grid operators. As of the intraday fluctuations, they may not require utility-scale, expensive batteries in the short term and mid-term because gas power plants can ramp up and down rapidly. Hence, even in the least restrictive scenario to hydropower expansion, we notice a shift from hydropower as main source of clean electricity to solar in the next years. This is akin to a paradigm shift in the power supply from a few, large infrastructure projects to multiple small decentralized power plants. This trend is in line with studies on other regions, for example, on Myanmar[44], Congo[28], or South Africa[25], and is robust against climate variability, as shown in our sensitivity analysis with different hydro-climatic conditions (see "Methods").

The key message of this paper is that the planned hydropower expansion should be revised in light of the new developments in the power market, in particular the fast decreasing costs of solar PV, which already produce the cheapest electricity in many countries[45]. Even in the Reference scenario, which reflects the Thai decarbonization targets and has no restrictions on hydropower portfolios, only 82% of the planned capacity is actually built according to our coupled models. If the power demand growth rates fall short of the projections, even less hydropower capacity would be needed. Thus, the construction of less economical dams should not proceed as planned. That being said, the Thai decarbonization targets until 2037 are not ambitious enough to push coal out of the system. In fact, we observe that more hydropower in the system enables coal to be used even more and comply with the $CO_2$ constraints. This trend applies not only to the Mekong countries but also to the whole ASEAN region. According to the International Energy Agency[1], the projected increase in fossil fuel consumption, particularly the

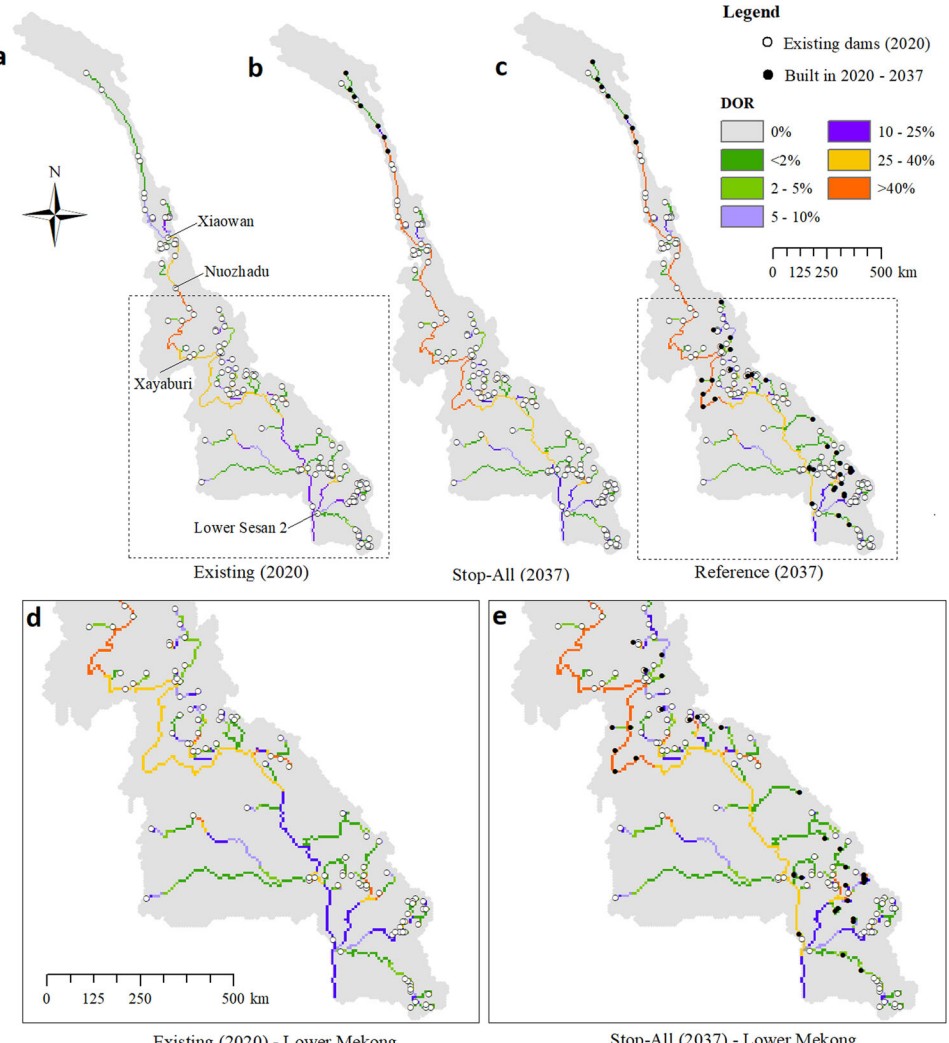

**Fig. 6 Effect of dams on river flow.** Change in the Degree of Regulation (DOR) between the current stage (2020) (**a**) and two dam development portfolios (Stop-All (**b**) and Reference (**c**)) in 2037. The two insets in (**d**, **e**) highlight the dam development conditions and DOR for the Lower Mekong Basin.

continued rise in coal demand, will lead to a two-thirds rise in $CO_2$ emissions and a 44% increase in premature deaths due to air pollution by 2040, compared to 2018. Therefore, revisions to policy plans have so far tended to boost the long-term share of renewable energy, typically at the expense of coal[1]. So ultimately, if decarbonization targets become more stringent in the long term or the mid-term, the competition between solar and hydro in the Mekong countries might turn into a collaboration, because they are probably both needed in large amounts to drive coal out.

Our study also demonstrates that there are technical pathways for combining the design of dam portfolios with the capacity expansion of power systems. By doing that, we can balance hydropower supply with environmental objectives and, importantly, explicitly evaluate the role played by dams within a power system, therefore avoiding the risk of conceiving portfolios that are economically or technically infeasible. By combining high-resolution hydrological and power system planning models, we also account for both geophysical and political boundaries, an information needed to account for limits and opportunities in cross-border power-trade infrastructure. Naturally, a modeling framework like ours should be used at the beginning, rather than at the end, of a conversation on sustainable energy planning, because its spatial domain and computational requirements

inevitably constrain the number of physical processes and scenarios that can be considered. In other words, screening models deployed across large domains should be complemented by local-scale impact assessments that evaluate additional, fundamental processes, such as sediment and fish passage through dams[46]. In this regard, another potential modeling avenue is to dynamically link strategic dam planning models and power system planning models, so as to provide a more exhaustive exploration of the ecology–energy trade-offs[44]. A local-scale assessment would also be more suitable for modeling extreme cases of demand fluctuations that test the reliability of the power system. Although this is not the top priority for developing countries that have not achieved universal access to electricity, it is safe to assume that the reliability requirements will soon converge toward the standards in the developed economies.

Looking forward, it is not difficult to imagine that many developing regions will be caught increasingly in the tension between ensuring cheap power security, exploiting locally available resources, and protecting ecosystems. Multi-model frameworks that span across multiple sectors—like the one described here—are a suitable platform for capturing these multiple perspectives and resolving, or at least addressing, ecology–energy trade-offs.

## Methods

**Hydrological and water management models**. To estimate the daily hydropower production of each dam in the Mekong and Chao Phraya basins, we adopt a two-step modeling approach. We begin with VIC, a large-scale, semi-distributed hydrologic model[47]. VIC organizes the spatial domain into a number of computational cells, where evapotranspiration, infiltration, baseflow, and runoff are calculated. The simulated runoff is then routed through the river network by VIC-Res, a water management model that includes an explicit representation of storage and release dynamics of water reservoirs[31]. In VIC-Res, each reservoir is represented by a cell accounting for dam location and a number of water of cells in which the storage dynamics are calculated. Daily release decisions are determined on the basis of bespoke rule curves. Using the information on hydraulic head and release, VIC-Res finally calculates the hydropower available at each dam.

Two separate computational domains were constructed to simulate hydrological and water management processes in the two basins. The domain for the Mekong covers an area of ~635,000 km$^2$, stretching from the Tibetan Plateau (China) to Kratie (Cambodia). In this model, we simulate the operations of 108 dams operational in 2016, spanning across China, Laos, Thailand, Cambodia, and Vietnam. This is necessary to account for the effect of upstream dam regulation on water availability and hydropower production in the Lower Mekong countries. The model for the Chao Phraya basin has a domain of ~110,000 km$^2$ and includes Bhumibol and Sirikit dams, which have a combined installed capacity of ~1280 MW. For both Mekong and Chao Phraya's models, we adopt a resolution of 1/16th of a degree, necessary to avoid allocating multiple dams to the same cell.

Key inputs include a Digital Elevation Model (DEM) and data on land use, soil, precipitation, and temperature. For the DEM, we masked the Global 30 Arc-Second Elevation (GTOPO30) DEM with the shape of the two basins and then adapted it to the resolution of our models with the average resampling technique[48]. Land use and soil data are obtained from the Global Land Cover Characterization dataset and Harmonized World Soil Database, respectively. The datasets have a spatial resolution of 30 arcsecond, so we generated land use and soil maps with the majority resampling technique. Rainfall and temperature data are retrieved from Global Meteorological Forcing Dataset[49], which have been thoroughly tested for our study site[50]. For the representation of reservoirs in VIC-Res, we acquired data on storage–depth relationship, maximum surface extent, dam design specifications, and rule curves. The storage–depth relationship is modeled with Liebe's method[51], the most common approach in large-scale studies[52–54]. The maximum surface extent of each reservoir is estimated by extracting surface water profiles from Landsat TM and ETM+ imagery, while the dam design specifications are obtained from the Mekong River Commission and the Electricity Generating Authority of Thailand—and complemented, where necessary, with information retrieved from other databases. Rule curves are designed to drawdown the reservoir storage during the driest months (December–May) to maximize the electricity production, recharge the depleted storage during the monsoon season, and avoid the risks of spilling water at the end of the monsoon season[30,55]. Rule curves are tailored to each reservoir by determining the time at which the minimum and maximum water levels are reached (roughly May and November), setting the value of the minimum and maximum water levels, and finally connecting these points with a piecewise linear function that gives us the daily target level for each calendar day[30]. With this modeling approach, rule curves account for both normal conditions and emergency procedures—that is, when the water level drops below the dead level or is above the critical one, requiring the activation of the spillways. Additional information on the input data is provided in Table S2.

To calibrate the hydrological model, we tuned the parameters controlling the rainfall-runoff process and compared the simulated discharge against the one observed at multiple gauging stations in the Mekong and Chao Phraya basins (data retrieved from the Mekong River Commission and the Thai Royal Irrigation Department). The calibration period is 1996–2005, with 1995 used for the model spin-up. During the simulation, reservoirs are activated in the year they become operational, so as to account for the non-stationarity of human interventions in the river basin. The model is then run over the period 2007–2016 (2006 is used for the spin-up). This validation includes a thorough comparison of the mean annual (simulated) hydropower production against the annual design (or expected) production. This is necessary to ensure that the rule curves correctly capture other factors affecting reservoir operations (e.g., irrigation) and therefore the hydropower profiles fed to the capacity expansion model. A detailed description of calibration and validation exercises is reported in refs. [22,23].

**Hydropower profiles for the capacity expansion model**. Ideally, the capacity expansion model should use as input a few years of hydropower profiles (simulated by VIC-Res for each dam), so as to explicitly account for the effect of inter-annual hydro-climatic variability. However, the computational requirements of the capacity expansion model prevent us from using multi-year profiles on a multi-regional model with hourly resolution, so we selected 2015 as a representative, or average, year. The effect of hydro-climatic variability on the hydropower profiles is illustrated in Figs. S3 and S4. As we shall see later, this variability has a marginal effect on the capacity expansion plans.

While VIC-Res provides a detailed accounting of the hydropower profiles for all dams built and operated in 2016, the capacity expansion model also needs hydropower profiles for dams planned over the period 2020–2037. To produce

them, we proceeded in two steps. First, we gathered information on location and design specifications of all planned dams (data provided by the Mekong River Commission and the Electricity Generating Authority of Thailand) and then added them to the power fleet simulated by VIC-Res. Specifically, we added the dams built over the period 2017–2019 (four in the Mekong, including Xayaburi Dam, and one in the Chao Phraya) and re-run VIC-Res with the same hydro-meteorological conditions used for the 2016's fleet. To determine the hydropower profile of the remaining dams (under construction in 2020 or at different planning stages in 2020–2037), we resorted to a proximity search—given the coordinates and installed capacity of a planned dam, we identify the most similar existing dam, from which the planned dam inherits the hydropower profile (see Fig. S5). This modeling choice is compelled by the absence of detailed design specifications (e.g., rule curves, maximum surface extent) needed to simulate planned dams with VIC-Res.

**Capacity expansion model**. We use the open-source modeling framework urbs to generate the model for the Lower Mekong countries. The model co-optimizes capacity expansion as well as hourly dispatch of generation, transmission, and storage from a social planner perspective. The goal of the optimization is to minimize the costs of expanding and operating the energy system, which include the annualized investment costs, fuel costs, and fixed and variable operations and maintenance costs. urbs solves a linear optimization problem that is written in Python/Pyomo using Gurobi.

Major inputs are the projected hourly electricity demand, hourly generation profiles of renewable energy technologies, the existing power infrastructure (power plants, grid, storage), planned expansion projects, emissions reduction targets, and techno-economic parameters, such as investment and maintenance costs, fuels costs, and specific emissions. Major outputs include the new capacities (generation, grid, storage) and the hourly operation of the system. The model also provides the direct emissions, the total costs, and the marginal electricity costs in each region.

The model has an hourly temporal resolution and models the years 2016 (the most recent year with comprehensive data availability), 2020, 2025, 2030, 2035, and 2037, for which the energy system targets of Thailand are defined. Assumptions about existing power plants, transmission lines, and techno-economic parameters are retrieved from the reports of the power system operators of the three countries, wherever possible. Missing data are completed from global sources. An overview of the data sources is available in Table S3.

For the year 2016, the hourly values of electricity demand in each province are obtained starting from province-wise, monthly varied peak electricity demand, collected from refs. [56–58]. The temporal disaggregation (from monthly to hourly values) is based on weekday/weekend and peak/off-peak demand profiles to account for the variation among days in a week and hours in a day. For the remaining years (i.e., 2020, 2025, 2030, 2035, and 2037), the hourly electricity demand in each province is obtained by scaling the 2016 profiles according to yearly demand growth projections for Thailand, Laos, and Cambodia[59–61].

As explained in the previous section, the hydropower profiles are obtained from VIC-Res. Because the model runs with a daily time step, we assumed that the hydropower profiles are uniformly available to urbs throughout 24 h. This input is derived using a single (representative) year. In a sensitivity analysis, we tested the impact of dry and wet conditions on the capacity expansion plans. In Fig. S6, we show that the plans are marginally affected by the hydro-climatic variability affecting the region (Figs. S3 and S4).

The model outputs of each year (new capacities) are used as inputs for the next one, to reflect short-sightedness in investment decisions. Regarding the spatial resolution, we use the provinces of Cambodia, Laos, and Thailand (25, 18, and 77, respectively) as model regions. Power demand and renewable generation time series are assigned to each region, as well as existing and planned power plant and storage capacities. Electricity transfer between the regions is allowed within the limits of the transmission capacities between them. Imports and exports to neighboring regions in China, Malaysia, Myanmar, and Vietnam are also constrained by the transmission capacities. The model assumes full coordination between Thailand, Laos, and Cambodia in the operation of the power grid. Trade with other countries stays within current levels, i.e., we do not consider that some of the planned dams will sell electricity to other markets in the ASEAN region.

The particularity of the model resides in the high level of spatial detail. The 120 model regions are small enough to preserve transmission bottlenecks and reflect their expansion costs without jeopardizing the model solvability. Within each region, there are different classes of solar and wind sites based on their potential energy output. Each class is characterized by a time series and an upper expansion capacity limit that reflect the quality and the availability of resources in the model region. Hence, the expansion of solar and wind power is solely based on their cost-competitiveness and not on exogenous expansion quotas. The maximum installable capacities of solar PV and onshore wind are 106.6 and 21.2 GW, respectively. The former value is equivalent to the most conservative estimation from a previous study[15], and the latter is obtained by applying a similar method. Whereas most power plant capacities are aggregated at the level of a model region, hydropower plants are modeled at the dam level, to avoid any information loss due to aggregation. Despite the higher computational burden, we made this modeling choice in line with the objectives of this study.

No system expansion is allowed in 2016, which is used only for calibration and validation (see Fig. S7). We compare the model performance against the

projections of the Power Development Plan of Thailand 2018–2037[59] (see Fig. S8). We observe minor differences that we are able to explain, and we conclude that the deviations do not affect the main conclusions of this paper.

**River fragmentation and regulation indices.** The RFI measures the loss of longitudinal connectivity in a river basin caused by hydraulic infrastructure. The RFI is defined as follows[33]:

$$\text{RFI} = 100 - \left( \sum_{i=1}^{n} \frac{v_i^2}{V^2} \cdot 100 \right), \quad (1)$$

where $n$ is the number of fragments (i.e., river network sections disconnected by dams), $v_i$ the volume of the $i$th fragment, and $V$ the total river volume (for the entire network). The RFI of a pristine river is 0%, while the one of a totally disconnected river is 100%. The impact of an individual dam depends on its location as well as the location of other dams. For example, a dam splitting a pristine network into two, equally sized fragments (in terms of volume) would change the RFI from 0 to 50%, but the construction of new dams close to this one would have smaller impact on the RFI[33]. A second important feature of the RFI is that it implicitly accounts for the larger impact of dams on the main stem and large tributaries by using the ratio between $v_i^2$ and $V^2$ as a weighting factor—since the river volume typically increases downstream due to increasing discharge and channel dimensions[32]. Following refs. [32,33], we assume that dams fully compromise connectivity and passability, that is, migrating fish and other species cannot move across two sections disconnected by a dam.

The RRI quantifies how strongly a river's hydrological regime is altered by dam operations. The RRI builds on the DOR, first introduced by ref. [35], which calculates, for each river reach, the discharge volume that can be withheld by a reservoir (or a group of reservoirs) located upstream. For a given reach, the DOR is defined as[32]:

$$\text{DOR} = \frac{\sum_{i=1}^{n} s_i}{D} \cdot 100, \quad (2)$$

where $n$ is the number of dams upstream of the reach, $s_i$ the storage capacity of the $i$th dam, and $D$ the total annual discharge volume of the river reach at hand. Large values of the DOR indicate that a substantial fraction of the discharge volume can be regulated by upstream storages, thereby increasing the chances of anthropogenic effects on the natural flow regime. The RRI is then calculated by weighting the DOR value of each individual reach with its corresponding river volume and then aggregating the results for the entire basin[32]:

$$\text{RRI} = \sum_{i=1}^{n} \text{DOR}_i \cdot \frac{v_i}{V}, \quad (3)$$

where $n$ is the total number of reaches, $\text{DOR}_i$ the DOR value of the $i$th reach, $v_i$ the corresponding volume, and $V$ the total river volume. Note that for a basin affected by multi-year, or carryover, reservoirs, the RRI value can be >100%.

The river network used for the calculation of the indices is based on VIC-Res flow direction matrix, which in turn is derived from the GTOPO30 DEM. Each cell of the matrix has a spatial resolution of 1/16th of a degree (roughly 7 km at the equator), resulting in a total of approximately 100,000 km of river network and 14,214 reaches. Following refs. [32,33], we then estimate the volume of each reach on the basis of average discharge (simulated by VIC-Res over the period 1986–2016) and an approximation of channel width and depth[62]. The information on storage capacity and location of each dam is retrieved from the same databases used to set up VIC-Res.

## Data availability
The data generated in this study have been deposited in Zenodo under the accession code https://doi.org/10.5281/zenodo.4837498.

## Code availability
The version of urbs used for this publication is available as an open source software in Zenodo with the identifier https://doi.org/10.5281/zenodo.4837475. VIC-Res is available at https://github.com/thanhiwer/VICRes.

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

## Acknowledgements

This research is supported by Singapore's Ministry of Education (MoE) through the Tier 2 project "Linking water availability to hydropower supply—an engineering systems approach" (Award No. MOE2017-T2-1-143). This work was also financially supported by Singapore's National Research Foundation under its Campus for Research Excellence And Technological Enterprise (CREATE) program.

## Author contributions

K.S. and S.G. designed the research. K.S. developed the capacity expansion model and led the analysis. T.D.D. developed the hydrological model and carried out the river fragmentation analysis. K.S., T.D.D., and A.F.M.K.C. prepared all simulation scenarios used for this study. K.S. and S.G. led the preparation of the manuscript, with substantive revision by all authors.

## Competing interests

The authors declare no competing interests.
