## [Peer Review File · Nature Communications]

REVIEWER COMMENTS

Reviewer #1 (Remarks to the Author):

This is an interesting analysis of the potential for improved electric power system design, facilitated by a highly resolved hydrologic-power systems model, that could substantially reduce the amount of hydropower capacity required to meet regional demands in Southeast Asia. While the analytical methodology does not appear entirely new, the interdisciplinary nature of the model and the high level of spatial resolution seem to be the critical elements that allow for new insights, although perhaps the temporal resolution also contributes to this as well (but is not entirely clear to me, see comment below). I find the paper interesting and with the potential to make a nice contribution to the literature, but I do have several questions, comments and requests for clarification.

This manuscript makes a good case that the alternative plans explored by the authors have the potential to meet regional demands, but I wonder about the assumptions in the model and the benchmarks used to evaluate the performance of the system. For example, I have some questions regarding the time step in modeled electricity demand, it is not clear, as there is mention of both a daily and hourly timesteps being used in the model. If its modeled at a daily time step, is there one daily profile for electricity demand or several (e.g., seasonal, weekday/weekend)? This would seem to be important given the huge role projected for solar energy, as the hourly (or less) intermittency issue will loom large. I also wonder whether the model has an established daily profile of demand, as opposed to developing a variable profile via a stochastic approach, perhaps actuated by temperature. I feel like the latter would be needed to evaluate system performance relative to the very high (six sigma) supply reliability that is sought in highly developed economies (or is a different standard applied here?). Also, is there any consideration of spatial correlations in demand highs/lows as a result of covariance in temperature/weather conditions across the region? I have similar questions related to the resolution with which the transmission system is modeled, particularly given the relatively large distance (and the number of nodes) that separate much of the hydropower generation and the primary demand centers. Any bottlenecks in this complex system would similarly have impacts on reliability.

Other comments/questions:

- I am confused by Figure 2. It appears to be plotting both countries and generation types, but I am not sure it does both. And if it does, I think it is not very clear. If the intent is to plot both sets of results, why not make the colors represent generation type and some pattern represent the country (i.e. dots, crosshatching, etc.).
- Are the "bespoke" guide curves used to simulate dam operations seasonal distinguished by monthly or weekly increments? Are there emergency procedures for high/low flow conditions, particularly the latter as these conditions could jeopardize power generation.
- Why is coal-based generation constant over all planning scenarios? It would seem to me that replacing coal- with natural gas-based generation over time would be in line with what is being observed in most regions. I would also expect that increased hydropower generation would not substitute for solar and/or natural gas based generation, but rather that it would replace coal.
- I do not fully understand the River Regulation Index (RRI), nor what it is so important that it needs to be given a prominent place in the text and the figures as it appears mostly invariant across all of the planning strategies explored at each time horizon.

Reviewer #2 (Remarks to the Author):

The manuscript by Siala et al. aims to highlight the opportunity that arise from solar power for replacing high-carbon coal fired plants and hydropower dams with impacts on rivers in the Mekong region. This is an important contribution, because it provides the most detailed analysis of energy

systems for the region thus far. The manuscript considers for many important aspects such as the spatial distribution of renewable resources, reservoir operation and hydrologic variability, and transmission, within the framework of an energy systems analysis. This is novel and has the potential for a broad range and real-world impact, given the current debates around energy in the basin. This manuscript is very suitable for Nature Communications after minor revisions.

Comments by line:

Title: The title highlights the 'battery of Asia' plan, which I don't think is too relevant for the framing. First, there is not too much mention of the battery of Asia throughout the manuscript. Second, the 'battery of Asia' concept is not an official plan but more a description of Laos' ambition to build a lot of hydropower. Hydropower still sees an important increase in many of the scenarios in the manuscript, which would still make Laos somewhat of a 'battery of Asia'. Lastly, the 'battery of Asia' is focussed on Laos, only, while a main strength of the paper is the regional perspective. Thus, I would recommend an alternative title. (E.g., Solar energy and regional coordination as a feasible alternative to large hydropower in South-East Asia)

The abstract could spend few sentences on making clearer what was actually done (i.e., a short highlight on the methods).

Lines 10 – 13: Consider adding some numbers. E.g., on the small cost differences of portfolios and the major difference in fragmentation.

Line 57 'information available from dam planning studies' is a bit ambivalent. Could be reformulated to 'information on environmental externalities available from dam planning studies'.

Line 94: costs for hydro are derived from global averages, or from cost estimates for each dam?

Line 97 – 98: There might be not an easy way for quantifying this, but it might be worth mentioning that projected growth rates are often inflated and very uncertain. Thailand also currently has a huge reserve margin. Is that considered to remain constant in the future?

Line 99: Is it necessary to introduce the two pathways as the two only options? Aren't there in theory a lot of different options spanning the continuum between the two pathways?

Line 153: 'stop main' might be a more intuitive abbreviation than 'stop stem'.

Line 157 – 159: This is an interesting observation. However, with global emissions markets and prices more close to the social cost of carbon, minimizing total emissions might become an additional objective for countries, making coal less attractive.

Line 166: It should be mentioned that many dams have major cost overruns which might make up for the small nominal cost advantage of hydro-heavy scenarios.

Line 212 – 214: It should be noted that some of the tributaries in the western part of the basin are regulated for irrigation, which does not show up in most hydropower-focused studies.

Line 394: Where and how are transmission costs considered?

Some of the figures are a bit unwieldy and crowded:

Figure S1 is very interesting as it shows the spatial distribution of the model, energy assets, and demands. Consider adding to main body?

Figure 2: Showing energy fluxes between countries and by technology would be interesting. Possible

extra figure for the SI?

Figure 4: Number of dams and hydropower indicator. The different sizes of hydropower markers are hard to distinguish and the second y axis makes interpretation difficult. I would recommend removing those and focusing on ROR, RFI only.

Figure 5: The inset box overlapping the main figure is not ideal. Consider showing the insets in a second row?

Panel references are inconsistent. Some figures use letters and titles (Fig 1), some figures have no panel labels (Figure 3, 5). I would recommend using letters and making clear reference to them in the captions.

Reply to reviewers of manuscript
NCOMMS-21-10540:
**Solar energy and regional coordination
as a feasible alternative to
large hydropower in Southeast Asia**

Kais Siala

TUM CREATE Ltd., Singapore

AFM Kamal Chowdhury

Environmental Studies Department,

University of California Santa Barbara, Santa Barbara, California

Pillar of Engineering Systems and Design

Singapore University of Technology and Design, Singapore

Thanh Duc Dang

Pillar of Engineering Systems and Design

Singapore University of Technology and Design, Singapore

Stefano Galelli

Pillar of Engineering Systems and Design

Singapore University of Technology and Design, Singapore

Email: stefano_galelli@sutd.edu.sg

Reviewer 1

This is an interesting analysis of the potential for improved electric power system design, facilitated by a highly resolved hydrologic-power systems model, that could substantially reduce the amount of hydropower capacity required to meet regional demands in Southeast Asia. While the analytical methodology does not appear entirely new, the interdisciplinary nature of the model and the high level of spatial resolution seem to be the critical elements that allow for new insights, although perhaps the temporal resolution also contributes to this as well (but is not entirely clear to me, see comment below). I find the paper interesting and with the potential to make a nice contribution the literature, but I do have several questions, comments and requests for clarification.

Thanks for your positive feedback and constructive comments, which helped us improve the quality of our work. In this revised version of the manuscript we provided more details on (1) the temporal resolution of the electricity demand and its relation to weather conditions, (2) the representation of the transmission network, and (3) the rule curves used to control hydropower reservoirs. In addition, we (4) improved the quality of Figure 2 and (5) clarified a few other points (e.g., evolution of coal-based generation over time, importance of the River Regulation Index).

Finally, please note that in our reply-to-the-reviewers line numbers refer to the track-and-changes version of the manuscript.

This manuscript makes a good case that the alternative plans explored by the authors have the potential to meet regional demands, but I wonder about the assumptions in the model and the benchmarks used to evaluate the performance of the system. For example, I have some questions regarding the time step in modeled electricity demand, it is not clear, as there is mention of both a daily and hourly timesteps being used in the model. If its modeled at a daily time step, is there one daily profile for electricity demand or several (e.g., seasonal, weekday/weekend)? This would seem to be important given the huge role projected for solar energy, as the hourly (or less) intermittency issue will loom large.

The capacity expansion model, urbs, uses hourly values of electricity demand for each province. For the year 2016—the first year of our simulation horizon—the demand data are obtained starting from province-wise, monthly varied peak electricity demand, collected from the 2016’s annual reports of EDC (Cambodia), EDL (Laos), and EGAT (Thailand). The temporal disaggregation (from monthly to hourly values) is based on weekday/weekend and peak/off-peak demand profiles to account for the variation among days in a week and hours in a day. The electricity demand data so obtained have been thoroughly evaluated in three previous studies on the Thai, Laotian, and Cambodian power systems (Chowdhury et al., 2020a, 2020b, 2021). For the remaining years (i.e., 2020,

2025, 2030, 2035, and 2037) the hourly electricity demand in each province is obtained by scaling the 2016 profiles according to yearly demand growth projections for Thailand (EPPO 2018), Laos (ADB 2019), and Cambodia (ADB 2018). We provided a more in-depth description of the demand data at Line 422-429. Regarding the solar and wind profiles, they are already generated with an hourly resolution using pyGRETA (<https://github.com/kais-siala/pyGRETA>).

The hydrological model, VIC-Res, uses a daily time step, so this potentially creates a mismatch between the two models. To solve it, we assume that the hydropower profiles (calculated by VIC-Res and used as input to urbs) are uniformly available throughout 24 hours. We clarified this point at Line 431-432.

The following table summarizes the explanations of the previous paragraphs.

Table 1: Original resolution versus model resolution for the demand, solar, wind, and hydropower profiles.

Data	Original resolution	Model resolution
Demand	Yearly (total demand by country), monthly (peak demand in each province), and hourly (weekly profiles including weekday/weekend and peak/off-peak variations)	Hourly
Solar and wind profiles	Hourly	Hourly
Hydropower profiles	Daily	Hourly

I also wonder whether the model has an established daily profile of demand, as opposed to developing a variable profile via a stochastic approach, perhaps actuated by temperature. I feel like the latter would be needed to evaluate system performance relative to the very high (six sigma) supply reliability that is sought in highly developed economies (or is a different standard applied here?). Also, is there any consideration of spatial correlations in demand highs/lows as a result of covariance in temperature/weather conditions across the region?

As mentioned in the answer to the previous question, the demand profile is a generic one that reflects weekday/weekend and peak/off-peak variations, monthly peak demand for each province, and yearly total demand per country. The demand profile is not explicitly temperature-dependent, but intra-annual temperature variations are implicitly reflected in the monthly peak demand of each province (the higher the temperature, the higher the peak because of air-conditioning). As shown in Figure 1 for the case of Thailand, the demand tends

to increase in the pre-monsoon months (which are drier and slightly warmer) and then decrease during the monsoon period.

Figure 1: Peak demand in Thailand (upper panel) and air temperature (lower panel) for the year 2016. The air temperature is measured at Bangna station, near Bangkok. The shaded area illustrates the range of variability across the years 2007-2019.

What our data and model setup do not capture is the inter-annual variability of demand (as a function of the meteorological conditions). We believe that this aspect is not an impediment to our analysis, which focuses mainly on the impact of different hydro expansion portfolios. When we investigated the effect of the hydro-climatic variability, we put an emphasis on the power supply, similarly to other studies such as Liu et al. (2017) and O’Connell et al. (2019). If we included the temperature-demand correlation for the scenarios with different hydrological conditions, it would probably lead to more solar PV installation in the Dry scenario (because solar PV correlates with the air-conditioning demand, and hydropower generation is less competitive in that scenario). The effect would then probably be in line with our current results.

To your question whether there is any consideration of spatial correlations in demand highs/lows as a result of covariance in temperature/weather conditions across the region, we believe that this is also covered by the monthly peak demand data on a provincial level. Regarding supply reliability considerations,

they are indeed important for any expansion planning exercise. Designing the system using the peak demand requires building supply capacities for an extreme case, and these capacities will probably be under-utilized. Tightening the supply reliability constraints even further will increase the installed capacity of dispatchable power plants (gas, coal, bioenergy), which would rarely operate. This is unlikely to happen in the region, particularly in Cambodia and Laos, which have not reached universal electricity access yet, and would probably opt for cheaper demand-side management options such as load shedding.

I have similar questions related to the resolution with which the transmission system is modeled, particularly given the relatively large distance (and the number of nodes) that separate much of the hydropower generation and the primary demand centers. Any bottlenecks in this complex system would similarly have impacts on reliability.

We used maps of high voltage transmission networks for the three countries (plus interconnections to their neighbors) to obtain our “simplified” transmission system. The aggregation method consists of adding the transmission capacities of the lines connecting any pair of regions directly. Lines that start and end in the same region are ignored, because we assume that each region behaves like a copper-plate where supply and load are connected with sufficient capacity. This way of modeling the transmission grid is usually referred to as a transport model (Schmid and Knopf (2015) and Schaber et al. (2012)). Recently, Cao et al. (2021) have shown that this simplification of the complex transmission network does not necessarily lead to an underestimation of the expansion costs in energy system optimization models, after comparing the method to more sophisticated approaches (direct current power flow model and linearized alternating current power flow model).

Figure 2 contrasts the original raw data with the aggregated grid. The original number of nodes within the three countries is 529, which include 360 substations (to which load time series are assigned) and 169 large power plants. We reduced it to 120 nodes (i.e., model regions), to which we associated the load and the power plants capacities that fall within the geographic boundaries of each of the administrative provinces. The reduction is necessary to ensure the tractability of the optimization problem considering the available computational capacity (using up to 36 CPU cores at 3.2GHz and 512 GB RAM, the model requires a couple of hours per modeled year and scenario).

As shown in Figure 2, thin, parallel lines are aggregated together in thicker corridors with a capacity equivalent to the sum of the individual line capacities. Similarly, regions with a sparse grid like Northern Cambodia or Western Thailand remain poorly connected in the equivalent grid. Hence, grid bottlenecks are reflected. In order to expand the grid, the model takes into account typical length-dependent cost assumptions per unit of power from the literature.

Figure 2: Original and simplified grids for the Mekong region in 2016.

Please refer to Figure 3 (d) (main manuscript) to find the added transmission capacities between 2020 and 2037.

Other comments/questions:

- *I am confused by Figure 2. It appears to be plotting both countries and generation types, but I am not sure it does both. And if it does, I think it is not very clear. If the intent is to plot both sets of results, why not make the colors represent generation type and some pattern represent the country (i.e. dots, crosshatching, etc.).*

Thank you for your feedback. To avoid any confusion, we decided to simplify the plot by grouping all imports and exports in one category, instead of listing the countries separately. Most of them were barely visible anyway, and it is now clear from the updated figure that most of the trade happens between Laos (net exporter) and Thailand (net importer). For your convenience, Figure 3 shows the updated graph.

- *Are the “bespoke” guide curves used to simulate dam operations seasonal distinguished by monthly or weekly increments? Are there emergency procedures for high/low flow conditions, particularly the latter as these conditions could jeopardize power generation.*

Figure 3: Capacity expansion plans for the period 2016–2037. Evolution of the power mix in Thailand, Laos, and Cambodia designed by urbs. Negative values indicate electricity exports that electricity exports exceed imports. Note that most of the electricity exported from Laos goes to Thailand.

Rule, or guide, curves are designed to drawdown the reservoir storage during the driest months (e.g., December to May) to maximize the electricity production, recharge the depleted storage during the monsoon season, and avoid the risks of spilling water at the end of the monsoon season (see Figure 4). Rule curves are tailored to each reservoir by determining the time at which the minimum and maximum water levels are reached (May and November, respectively), setting the value of the minimum and maximum water levels, and finally connecting these points with a piecewise linear function that gives us the daily target level for each calendar day—meaning that release decisions are made with daily increments.

To answer the second question (on emergency procedures for high/low flow conditions), we resort again to Figure 4, where the numbers (1)-(4) indicate the four storage zones in which a reservoir is organized. Each zone is associated to different release decisions. If the level falls in Zone 1 (below the dead water level), water is not released. If the level is in Zone 2 (between dead and target levels), VIC-Res uses information on the incoming daily inflow to calculate whether the water level is expected to exceed the target one by the end of the day. If that is the case, VIC-Res discharges the amount of water needed to keep the actual level as close as possible to the target. Otherwise, water is not

Figure 4: Illustration of a rule curve. The parameters T_1 and T_2 represent the time at which the minimum and maximum water levels are reached, while H_1 and H_2 are the corresponding (minimum and maximum) water levels. The minimum and maximum elevation levels are denoted with H_{\min} and H_{\max} . The numbers (1)-(4) denote the four storage zones in which a reservoir is organized.

discharged. In Zone 3 (between the target and critical levels), water is released using the dam design discharge, so as to quickly reach the target level. When the level reaches Zone 4 (beyond the critical level), spillways are activated, so as to avoid overtopping. Zone (1) and (4) therefore account for the emergency conditions encountered during periods of low and high flows, respectively.

We understand that both aspects (i.e., temporal resolution of the rule curves and emergency procedures) were not explained clearly, so we proceeded to revise them at Line 352-362.

- *Why is coal-based generation constant over all planning scenarios? It would seem to me that replacing coal- with natural gas-based generation over time would be in line with what is being observed in most regions. I would also expect that increased hydropower generation would not substitute for solar and/or natural gas based generation, but rather that it would replace coal.*

In Figure 4 of the manuscript, we observe that the coal generation is constant in the first years across the scenarios, but varies after 2035 (highest in Reference at 54 TWh, lowest in Stop-All at 41 TWh). The evolution of coal generation in

the model is indeed counter-intuitive if compared with its evolution in Europe or North America, for example, but is actually in line with other studies for Southeast Asia (IEA 2019). The main difference between the regions lies in the decarbonization policies: in Europe, the EU-ETS drives the cost of coal generation upwards at a faster rate than gas because coal generation emits more CO₂, making it less competitive. In North America, shale gas is cheap and is driving coal out of the market. In the Mekong countries, particularly in Laos, coal is abundant and cheap. To our knowledge, there is currently no carbon-tax system, and is considering a carbon tax as part of a set of policy measures to reduce its emissions (IEA 2020), but its plans are still vague. Therefore, we only modeled its commitment to reduce its carbon intensity (and extrapolated it to Laos and Cambodia). Taking into account the growing electricity demand in the region, the reduction of the carbon intensity per unit of energy could happen by adding cleaner sources of energy without reducing the amount of coal drastically—in fact, the amount of coal used may increase, as seen in the Reference scenario, when there is abundant hydro generation, which is carbon-free. By reflecting the current policies and objectives, the model results serve as a warning that the decarbonization targets in the region are rather unambitious (compared to developed economies) and insufficient to drive out coal (which does not seem to be a target based on the current policies in these countries).

- *I do not fully understand the River Regulation Index (RRI), nor what it is so important that it needs to be given a prominent place in the text and the figures as it appears mostly invariant across all of the planning strategies explored at each time horizon.*

Dams have two major impacts on riverine systems: (1) they disconnect the river network into a number of fragments, or sections, (2) they alter the hydrological regime, that is, they change the natural timing and magnitude of flows. In general, hydropower dams tend to increase the minimum flows (to sustain power production) and decrease peak flows (because dams tend to store water and use it during dry periods). Both aspects (fragmentation of the river network and alteration of the hydrological regime) are important, because they have different effects on freshwater species. In our case, this means it is important to account for both River Fragmentation Index (RFI) and River Regulation Index (RRI)—as also done in previous studies (e.g., Grill et al., 2014).

This being said, we agree with the reviewer that it is somewhat surprising to see so much emphasis on an indicator (RRI) that is pretty much constant over the period 2020–2035 (there is an increase for the year 2037). So, why keeping it? The answer, in our opinion, lies in the role of the Chinese dams. Our analysis shows that the Lower Mekong countries (Thailand, Laos, and Cambodia) have tangible opportunities for replacing planned dams with solar PV and other energy sources. Such option, represented by a few different dam portfolios,

has a clear impact on the river fragmentation (RFI), but almost no impact on the hydrological regime. The reason is that the majority of the planned dams are run-off-river, with limited storage capacity relative to inflow. The river regulation is mostly affected by existing and planned dams in the Upper Mekong (China), which have massive storage capacity relative to inflow. In synthesis, our analysis indicates that river fragmentation can be largely tackled by the Lower Mekong countries, but ensuring that the timing and magnitude of flows do not deviate too much from natural conditions requires the cooperation of all countries, including China. We tried to further emphasize this point in the revised version of the manuscript (Line 203-207).

Reviewer 2

The manuscript by Siala et al. aims to highlight the opportunity that arise from solar power for replacing high-carbon coal fired plants and hydropower dams with impacts on rivers in the Mekong region. This is an important contribution, because it provides the most detailed analysis of energy systems for the region thus far. The manuscript considers for many important aspects such as the spatial distribution of renewable resources, reservoir operation and hydrologic variability, and transmission, within the framework of an energy systems analysis. This is novel and has the potential for a broad range and real-world impact, given the current debates around energy in the basin. This manuscript is very suitable for Nature Communications after minor revisions.

Thanks for the positive feedback and many comments on how to improve our paper. We addressed all of them, paying particular attention to (1) the evolution of coal-based generation, (2) transmission costs, and (3) data visualization. Finally, please note that in our reply-to-the-reviewers line numbers refer to the track-and-changes version of the manuscript.

Comments by line:

- *Title: The title highlights the ‘battery of Asia’ plan, which I don’t think is too relevant for the framing. First, there is not too much mention of the battery of Asia throughout the manuscript. Second, the ‘battery of Asia’ concept is not an official plan but more a description of Laos’ ambition to build a lot of hydropower. Hydropower still sees an important increase in many of the scenarios in the manuscript, which would still make Laos somewhat of a ‘battery of Asia’. Lastly, the ‘battery of Asia’ is focussed on Laos, only, while a main strength of the paper is the regional perspective. Thus, I would recommend an alternative title. (E.g., Solar energy and regional coordination as a feasible alternative to large hydropower in South-East Asia)*

We agree with your suggestion and modified the title accordingly. We also updated the rest of the manuscript in a couple of instances (Line 4-5 and 87).

- *The abstract could spend few sentences on making clearer what was actually done (i.e., a short highlight on the methods).*

We agree with the suggestion. Regretfully, we are not able to implement it because of the word limit of 150 words. With such constraint, we prefer to focus on problem framing and key results.

- *Lines 10 – 13: Consider adding some numbers. E.g., on the small cost differences of portfolios and the major difference in fragmentation.*

Thanks for the suggestion. We added the percentage increase in costs (Line 12).

- *Line 57 ‘information available from dam planning studies’ is a bit ambivalent. Could be reformulated to ‘information on environmental externalities available from dam planning studies’.*

We modified the sentence as suggested.

- *Line 94: costs for hydro are derived from global averages, or from cost estimates for each dam?*

Since the hydropower expansion is critical for our analysis, we decided to use cost assumptions specific for the region (AGEP 2019). The data source is also referenced in the Supplementary Information in Table S3.

- *Line 97 – 98: There might be not an easy way for quantifying this, but it might be worth mentioning that projected growth rates are often inflated and very uncertain. Thailand also currently has a huge reserve margin. Is that considered to remain constant in the future?*

It is true that growth projections are highly uncertain and usually inflated. If the demand in the Mekong countries falls short of the projections, this would only strengthen our key message: rolling back part of the dam expansion plans is necessary in light of recent developments in solar technology costs. In a future with less power demand, even more dams would probably be redundant. We added a sentence in the manuscript to address this aspect (Lines 275–276).

Reserve margins are currently not considered in the model. We chose to design the system using data of peak demand (please refer to our response to reviewer #1, first comment), which already leads to building supply capacities for an extreme case, and these will probably be under-utilized. Tightening the constraints on reserve margins on top of that will increase the installed capacity of dispatchable power plants (gas, coal, bioenergy), which would rarely operate. This is unlikely to happen in the region, particularly in Cambodia and Laos, which have not reached universal electricity access yet, and would probably opt for cheaper demand-side management options such as load shedding. For Thailand, we are not assuming that the reserve margin will remain constant in the future.

- *Line 99: Is it necessary to introduce the two pathways as the two only options? Aren't there in theory a lot of different options spanning the continuum between the two pathways?*

We reformulated the sentence in Lines 105–109. It now reads: “There are many decarbonization pathways to reach these targets, but they roughly fit into two categories. The first one focuses on shifting from coal to gas (which has a lower carbon intensity), with a moderate expansion of renewable energy technologies. The second relies on a large expansion of renewable energy and a moderate expansion of gas power plants, so that the system can accommodate a continuous usage of coal.”

- *Line 153: ‘stop main’ might be a more intuitive abbreviation than ‘stop stem’.*

We modified the text, table, and figures as suggested.

- *Line 157 – 159: This is an interesting observation. However, with global emissions markets and prices more close to the social cost of carbon, minimizing total emissions might become an additional objective for countries, making coal less attractive.*

It is unclear whether developing economies like Laos, which have abundant coal resources, would be inclined to voluntarily reduce their use of coal—or whether they would be forced to be part of a global emissions market that reflects the social cost of carbon. There are currently no such markets, and the more realistic plans in the mid-term would involve developed economies first.

The projected continued usage of coal is actually in line with other studies for Southeast Asia (IEA 2019). To our knowledge, there is currently no carbon-tax in the analyzed Mekong countries. Thailand has started experimenting with a cap-and-trade system, and is considering a carbon tax as part of a set of policy measures to reduce its emissions (IEA 2020), but its plans are still vague. Therefore, we only modeled its commitment to reduce its carbon intensity (and extrapolated it to Laos and Cambodia). Taking into account the growing electricity demand in the region, the reduction of the carbon intensity per unit of energy could happen by adding cleaner sources of energy without reducing the amount of coal drastically—in fact, the amount of coal used may increase, as seen in the Reference scenario, when there is abundant hydro generation, which is carbon-free. By reflecting the current policies and objectives, the model results serve as a warning that the decarbonization targets in the region are rather unambitious (compared to developed economies) and insufficient to drive out coal (which does not seem to be a target based on the current policies in these countries).

- *Line 166: It should be mentioned that many dams have major cost overruns which might make up for the small nominal cost advantage of hydro-heavy scenarios.*

Good point; thanks for suggesting it. We now mention it in the first paragraph of Section 3 (Line 242-243), where we discuss about the economic feasibility of alternative dam portfolios. To support this point, we added a reference to Braeckman et al., 2020.

- *Line 212 – 214: It should be noted that some of the tributaries in the western part of the basin are regulated for irrigation, which does not show up in most hydropower-focused studies.*

In our hydrological model, VIC-Res, the operating rules are manually adjusted to ensure that the mean annual (simulated) hydropower production is similar to the annual design (or expected) production. By doing that, we implicitly account for other operating objectives, such as irrigation demand. This indeed important for a few reservoirs located in the western part of the Mekong basin as well as the two main reservoirs in the Chao Phraya basin (Bhumibol and Sirikit). We understand that this aspect was not reported with sufficient clarity in the first version of the manuscript, so we took this opportunity to clarify it in a couple of instances—Line 224-226 (as suggested by the reviewer) and Line 372-375 (“Methods / Hydrological and water management models”).

- *Line 394: Where and how are transmission costs considered?*

We used maps of high voltage transmission networks for the three countries (plus interconnections to their neighbors) to derive a “simplified” transmission system with fewer regions. The aggregation method consists of adding the transmission capacities of the lines connecting any pair of regions directly. Lines that start and end in the same region are ignored, because we assume that each region behaves like a copper-plate where supply and load are connected with sufficient capacity. This way of modeling the transmission grid is usually referred to as a transport model (Schmid and Knopf (2015) and Schaber et al. (2012)). Recently, Cao et al. (2021) have shown that this simplification of the complex transmission network does not necessarily lead to an underestimation of the expansion costs in energy system optimization models, after comparing the method to more sophisticated approaches (direct current power flow model and linearized alternating current power flow model). The reduction is necessary to ensure the tractability of the optimization problem considering the available computational capacity (using up to 36 CPU cores at 3.2GHz and 512 GB RAM, the model requires a couple of hours per modeled year and scenario).

For the interregional lines, we assume that they start and end in the centroids of the regions, and calculate their lengths using a GIS software. In order to expand the grid, the model takes into account typical length-dependent cost assumptions per unit of power from the literature. If a transmission line expansion leads to a cost-optimal solution, it will be built, similarly to power plants. Please refer to Figure 3 (d) (main manuscript) to find the added transmission capacities between 2020 and 2037.

- *Some of the figures are a bit unwieldy and crowded: Figure S1 is very interesting as it shows the spatial distribution of the model, energy assets, and demands. Consider adding to main body?*

Following your advice, we moved Figure S1 to the main body (currently Figure 3).

- *Figure 2: Showing energy fluxes between countries and by technology would be interesting. Possible extra figure for the SI?*

The previous version of Figure 2 used to show the energy fluxes between countries, but most of the values were insignificant, hence the bars did not appear in the plot. We decided to group the energy trade into one legend item, showing the balance (imports minus exports). We think that the current version is much clearer, particularly regarding the importance of the electricity trade between Laos and Thailand (visible in the bars of almost similar sizes, one corresponding to exports from Laos and the other to imports to Thailand). Differentiating the trade by technology is tricky, since the regions trade electricity, without specifying whether it comes from hydropower or from coal. There are ways to approximate the shares of technologies, for example by looking at the mix in every hour and assuming that it applies to the exports in that time step. However, this is a rough approximation which, in our opinion, has little added value to the message of the paper. Besides, it does not reflect the existing power purchase agreements (PPA) between countries. In fact, Thailand has many PPAs with hydropower dams in Laos, and many transmission lines between the two countries are designed to transmit the hydropower to the load centers in Thailand. Hence, even if the power mix of Laos has a large share of coal, it is safe to assume that the hydropower is driving the exports to Thailand.

- *Figure 4: Number of dams and hydropower indicator. The different sizes of hydropower markers are hard to distinguish and the second y axis makes interpretation difficult. I would recommend removing those and focusing on ROR, RFI only.*

We modified the figure as suggested.

- *Figure 5: The inset box overlapping the main figure is not ideal. Consider showing the insets in a second row?*

Modified as suggested; it looks better now.

- *Panel references are inconsistent. Some figures use letters and titles (Figure 1), some figures have no panel labels (Figure 3, 5). I would recommend using letters and making clear reference to them in the captions.*

We added the letters for the subfigures and mentioned them in the captions.

References

- ADB, 2018. Cambodia: Energy Sector Assessment, Strategy, and Road Map. Technical Report, Asian Development Bank (ADB), Manila, Philippines. URL: <https://www.adb.org/documents/cambodia-energy-assessment-strategy-road-map>.
- ADB, 2019. Lao People’s Democratic Republic: Energy Sector Assessment, Strategy, and Road Map. Technical Report, Asian Development Bank (ADB), Manila, Philippines. URL: <https://www.adb.org/documents/lao-pdr-energy-assessment-strategy-road-map>.
- AGEP, 2019. Levelised Costs of Electricity for Renewable Energy Technologies in ASEAN Member States II. Technical Report, ASEAN Centre for Energy (ACE), Jakarta, Indonesia. URL: <https://aseanenergy.org/levelised-costs-of-electricity-for-renewable-energy-technologies-in-asean-member-states-ii/>.
- Braeckman, J.P., Disselhoff, T. and Kirchherr, J., 2020. Cost and schedule overruns in large hydropower dams: An assessment of projects completed since 2000. *International Journal of Water Resources Development*, 36:5, 839-854.
- Cao, K.-K., Pregger, T., Haas, J., Lens, H., 2021. To Prevent or Promote Grid Expansion? Analyzing the Future Role of Power Transmission in the European Energy System. *Frontiers in Energy Research*, 8(2), 371. doi: 10.3389/fenrg.2020.541495.
- Chowdhury, A.K., Kern, J., Dang, T.D. and Galelli, S., 2020a. PowNet: a network-constrained unit commitment/economic dispatch model for large-scale power systems analysis. *Journal of Open Research Software*, 8(1).
- Chowdhury, A.K., Dang, T.D., Bagchi, A. and Galelli, S., 2020b. Expected Benefits of Laos’ Hydropower Development Curbed by Hydroclimatic Variability and Limited Transmission Capacity: Opportunities to Reform. *Journal of Water Resources Planning and Management*, 146(10), p.05020019.
- Chowdhury, A.K., Dang, T.D., Nguyen, H.T., Koh, R. and Galelli, S., 2021. The Greater Mekong’s Climate-Water-Energy Nexus: How ENSO-Triggered Regional Droughts Affect Power Supply and CO₂ Emissions. *Earth’s Future*, 9(3), p.e2020EF001814.
- EDC. Annual report 2016. Technical report, Electricite Du Cambodge (EDC), Phnom Penh, Cambodia, 2016.
- EDL. Annual report 2016. Technical report, Electricite Du Laos (EDL), Vientiane Capital, Lao PDR, 2016.

EGAT. Annual report 2016. Technical report, Electricity Generating Authority of Thailand (EGAT), Nonthaburi, Thailand, 2016.

EPPO. Thailand power development plan 2018–2037 (PDP2018). Technical report, Energy Policy and Planning Office (EPPO), Ministry of Energy, Thailand, 2018.

Grill, G., Dallaire, C. O., Chouinard, E. F., Sindorf, N., and Lehner, B., 2014. Development of new indicators to evaluate river fragmentation and flow regulation at large scales: A case study for the Mekong River Basin. *Ecological Indicators*, 45, 148–159.

IEA, 2019. Southeast Asia energy outlook 2019. International Energy Agency, Paris, France. URL: <https://www.iea.org/reports/southeast-asia-energy-outlook-2019>.

IEA, 2020. Putting a price on carbon – an efficient way for Thailand to meet its bold emission target. International Energy Agency, Paris, France. URL: <https://www.iea.org/articles/putting-a-price-on-carbon-an-efficient-way-for-thailand-to-meet-its-bold-emission-target>.

Liu, L., Hejazi, M., Li, H., Forman, B., and Zhang, X., 2017. Vulnerability of US thermoelectric power generation to climate change when incorporating state level environmental regulations. *Nature Energy*, 2 (8), 1–5.

O’Connell, Voisin, N., Macknick, and Fu, 2019. Sensitivity of Western U.S. power system dynamics to droughts compounded with fuel price variability. *Applied Energy*, 247, 745 – 754, doi: 10.1016/j.apenergy.2019.01.156.

Schaber, K., Steinke, F., and Hamacher, T., 2012. Transmission grid extensions for the integration of variable renewable energies in Europe: who benefits where? *Energy Pol.* 43, 123–135. doi: 10.1016/j.enpol.2011.12.040.

Schmid, E., and Knopf, B., 2015. Quantifying the long-term economic benefits of European electricity system integration. *Energy Pol.* 87, 260–269. doi: 10.1016/j.enpol.2015.09.026.

REVIEWERS' COMMENTS

Reviewer #1 (Remarks to the Author):

The authors have made a complete and good faith effort to address my comments on the original manuscript, and I am mostly satisfied with their response.

The one place in which I think it would be good for them to elaborate further would be on the limitations of their described approach if applied in a developing world context. The lack of a stochastic demand element seems limiting if this approach were applied in most developed countries as it is the extremes in demand, brought about by extreme temperatures, that are now the focus of most planning exercises. Assuming a roughly constant monthly profile for demand, even with consideration of the 95% confidence interval, is probably insufficient for developing systems that must meet very high levels of reliability. In these cases, all of the "interesting" (i.e. problematic) events take place in the tails of the demand distribution, well beyond the 95th percentile. While cost and other development considerations may change the benchmark goals for reliability in the countries considered in this work, the difference in framing would be something worth noting.

Reviewer #2 (Remarks to the Author):

The authors have addressed all comments very well, making the figures and some parts of the text much easier to follow. Few minor comments on Figure 1:

Panel b is entitled (electricity system as of 2016), this makes me wonder as of when the hydropower system is, i.e., which year does the dam data set reflect?

In the legend, what is the meaning of '5200' close to the pie chart for 'Existing capacity'? If you mean that the size of the pie scales with the total capacity, then this is not overly clear.

R. J. P. Schmitt

Reply to reviewers of manuscript
NCOMMS-21-10540A:
**Solar energy and regional coordination
as a feasible alternative to
large hydropower in Southeast Asia**

Kais Siala

TUM CREATE Ltd., Singapore

AFM Kamal Chowdhury

Environmental Studies Department,

University of California Santa Barbara, Santa Barbara, California

Pillar of Engineering Systems and Design

Singapore University of Technology and Design, Singapore

Thanh Duc Dang

Pillar of Engineering Systems and Design

Singapore University of Technology and Design, Singapore

Stefano Galelli

Pillar of Engineering Systems and Design

Singapore University of Technology and Design, Singapore

Email: stefano_galelli@sutd.edu.sg

Reviewer 1

The authors have made a complete and good faith effort to address my comments on the original manuscript, and I am mostly satisfied with their response.

Thanks for your positive feedback.

The one place in which I think it would be good for them to elaborate further would be on the limitations of their described approach if applied in a developing world context. The lack of a stochastic demand element seems limiting if this approach were applied in most developed countries as it is the extremes in demand, brought about by extreme temperatures, that are now the focus of most planning exercises. Assuming a roughly constant monthly profile for demand, even with consideration of the 95% confidence interval, is probably insufficient for developing systems that must meet very high levels of reliability. In these cases, all of the "interesting" (i.e. problematic) events take place in the tails of the demand distribution, well beyond the 95th percentile. While cost and other development considerations may change the benchmark goals for reliability in the countries considered in this work, the difference in framing would be something worth noting.

We elaborated on this point in the final part of the Discussion (Line 298–302), which now includes the following consideration: “A local-scale assessment would also be more suitable for modeling extreme cases of demand fluctuations that test the reliability of the power system. Although this is not the top priority for developing countries that have not achieved universal access to electricity, it is safe to assume that the reliability requirements will soon converge towards the standards in developed economies.”

Reviewer 2

The authors have addressed all comments very well, making the figures and some parts of the text much easier to follow. Few minor comments on Figure 1:

Dear Dr. Schmitt, thank you for your positive feedback. As explained below, we took this opportunity to clarify the last outstanding issues with Figure 1.

Panel b is entitled (electricity system as of 2016), this makes me wonder as of when the hydropower system is, i.e., which year does the dam data set reflect?

In Figure 1, both panels illustrate the infrastructure (hydropower on the left and entire power system on the right) built and operated in 2016. In the left

panel, we also report (with a different symbol) the planned dams—which are therefore considered in our capacity expansion exercise. We updated the title of panel b to clarify this point.

In the legend, what is the meaning of '5200' close to the pie chart for 'Existing capacity'? If you mean that the size of the pie scales with the total capacity, then this is not overly clear.

Yes, we mean that the size of the pie scales with the total capacity. We understand that this point was not totally clear, so we revised the caption and added “MW (size-scale)” next to “5200”.